# Timeliness of routine childhood vaccination in 103 low-and middle-income countries, 1978–2021: A scoping review to map measurement and methodological gaps

Oghenebrume Wariri [1,2,3]*, Uduak Okomo [1], Yakubu Kevin Kwarshak [4], Chigozie Edson Utazi [5,6], Kris Murray [7,8], Chris Grundy [2‡], Beate Kampmann [1,3‡]

**1** Vaccines and Immunity Theme, MRC Unit The Gambia at London School of Hygiene and Tropical Medicine, Fajara, The Gambia, **2** Department of Infectious Disease Epidemiology, London School of Hygiene and Tropical Medicine, London, United Kingdom, **3** Vaccine Centre, London School of Hygiene and Tropical Medicine, London, United Kingdom, **4** Department of Surgery, Jos University Teaching Hospital, Plateau State, Nigeria, **5** WorldPop, School of geography and Environmental Science, University of Southampton, Southampton, United Kingdom, **6** Southampton Statistical Sciences Research Institute, University of Southampton, Southampton, United Kingdom, **7** MRC Unit The Gambia at The London School of Hygiene and Tropical Medicine, Fajara, The Gambia, **8** MRC Centre for Global Infectious Disease Analysis, Imperial College School of Public Health, Imperial College London, London, United Kingdom

‡ CG and BK are joint senior authors on this work.
* Oghenebrume.Wariri@lshtm.ac.uk

**Data Availability Statement:** All related data are included in the mansucript or in the supplementary file.

## Abstract

Empiric studies exploring the timeliness of routine vaccination in low-and middle-income countries (LMICs) have gained momentum in the last decade. Nevertheless, there is emerging evidence suggesting that these studies have key measurement and methodological gaps that limit their comparability and utility. Hence, there is a need to identify, and document these gaps which could inform the design, conduct, and reporting of future research on the timeliness of vaccination. We synthesised the literature to determine the methodological and measurement gaps in the assessment of vaccination timeliness in LMICs. We searched five electronic databases for peer-reviewed articles in English and French that evaluated vaccination timeliness in LMICs, and were published between 01 January 1978, and 01 July 2021. Two reviewers independently screened titles and abstracts and reviewed full texts of relevant articles, following the guidance framework for scoping reviews by the Joanna Briggs Institute. From the 4263 titles identified, we included 224 articles from 103 countries. China (40), India (27), and Kenya (23) had the highest number of publications respectively. Of the three domains of timeliness, the most studied domain was 'delayed vaccination' [99.5% (223/224)], followed by 'early vaccination' [21.9% (49/224)], and 'untimely interval vaccination' [9% (20/224)]. Definitions for early (seven different definitions), untimely interval (four different definitions), and delayed vaccination (19 different definitions) varied across the studies. Most studies [72.3% (166/224)] operationalised vaccination timeliness as a categorical variable, compared to only 9.8% (22/224) of studies that operationalised timeliness as continuous variables. A large proportion of studies [47.8% (107/224)] excluded the data of children with no written vaccination records irrespective of caregivers' recall of

**Funding:** This project is part of the EDCTP2 Programme supported by the European and Developing Countries Clinical Trials Partnership (grant number TMA2019CDF-2734 - TIMELY). OW is also supported by an Imperial College London Wellcome Trust Institutional Strategic Support Fund (ISSF) (grant no RSRO_P67869). KM is supported by joint Centre funding from the UK Medical Research Council and Department for International Development [MR/R0156600/1]. CEU is supported by funding from the Bill & Melinda Gates Foundation (Investment ID INV-003287). The Vaccines and Immunity Theme (OW, UO, and BK) is jointly funded by the UK MRC and the UK Department for International Development (DFID) under the MRC/DFID Concordat agreement and is also part of the EDCTP2 Programme supported by the EU (MC UP_A900/1122, MC UP A900/115). The funders had no role in study design, data collection and analysis, decision to publish, or preparation of the manuscript.

**Competing interests:** The authors have declared that no competing interests exist

their vaccination status. Our findings show that studies on vaccination timeliness in LMICs has measurement and methodological gaps. We recommend the development and implement of guidelines for measuring and reporting vaccination timeliness to bridge these gaps.

## Introduction

Since its inception in 1974, the expanded programme on immunisation (EPI) has successfully decreased the incidence of, and mortality from childhood vaccine preventable diseases (VPDs), nevertheless, progress has plateaued, or regressed in many countries [1]. Vaccination coverage dropped globally by about 3% between 2019–2020, with an estimated 23 million children under the age of one year not receiving their basic vaccines in 2020 –the highest number since 2009 [2]. In addition, 8·9 million children were not routinely vaccinated with the first-dose measles-containing vaccine (MCV1) which prevent measles, a highly contagious infectious disease [3]. Disruptions to routine childhood vaccination due to the ongoing pandemic are likely to amplify the already existing gaps which prevented countries from reaching global immunisation targets [4].

The traditional metric used for evaluating the success of immunisation programs is vaccine-specific crude vaccination coverage [5]. Crude vaccination coverage conceptually assumes uptake of vaccines without considering timely delivery, i.e., whether doses are received within the recommended window, are too early, delayed, or whether the intervals between doses are inappropriate [6]. To achieve the full benefit of vaccines, however, both high coverage and timely delivery are required. Timeliness of vaccination–i.e., vaccination received within the recommended window, in an age-appropriate manner explores the quality dimension of immunisation programs and is important for several reasons. Untimely vaccination might be the only early warning sign that could alert EPI programme managers to potential problems with the delivery of certain vaccines, and help put in place mitigating strategies. Vaccines received too early, or before the earliest valid ages may result in suboptimal immunity due to interference with maternal antibodies [7]. Delayed vaccination, on the other hand, prolongs the exposure of children to debilitating VPDs such as *Haemophilus influenzae* type b, pertussis, and measles whose peaks occur in infancy [7, 8]. Delayed vaccination also increases a child's risk of not completing their schedule, and ultimately leads to suboptimal levels of herd immunity needed to prevent the outbreak of VPDs. There is evidence suggesting that measles outbreaks have occurred in the past due to delayed vaccination despite high overall crude vaccination coverage [9].

Over the last decade, studies exploring vaccination timeliness have gained some traction [10]. A recent Global Burden of Disease Study published in *The Lancet* argued that vaccination timeliness better reflects coverage trend, thus, recommended that future research should estimate age-specific vaccination coverage rather than crude coverage alone [11]. Most vaccination timeliness studies have been conducted in high-income countries (HICs) with much fewer reports from low-and middle-income countries (LMICs) where vaccination coverage is variable but comparatively lower, and VPD burden is high [10]. There is emerging evidence suggesting that the published studies on the timeliness of routine vaccination in LMICs has key methodological and measurement issues that limit their comparability, utility, and the extent to which inference can be drawn from their findings [10]. Hence, there is an urgent need to identify, and document these measurement and methodological gaps which could

inform the design, conduct, and reporting of future research on the timeliness of routine childhood vaccination in LMICs.

This scoping review, therefore, aimed to identify and synthesise published literature on the timeliness of routine childhood vaccination in LMIC and answer the following questions: (a) how has the literature on vaccination timeliness evolved?; (b) how has vaccination timeliness been defined or operationalisation?; (c) what domains of vaccination timeliness have been studied; (d) what methodological or statistical approaches have previous studies deployed to ensure robustness of results and; (e) what determinants of untimely vaccination have been explored.

## Materials and methods

Scoping reviews are an emerging approach for evidence synthesis. Unlike systematic reviews that traditionally answer precise questions related to the effectiveness of a specific intervention, scoping reviews are exploratory in nature [12]. Scoping reviews typically address a broad question such as what kind of evidence exists on a topic, and how research on that topic has been designed or conducted [13]. They are useful in mapping the key concepts underpinning a research area as well as to clarify working definitions or concepts, and identify knowledge gaps [12]. These characteristics make the scoping review approach well suited to answer our research questions aimed at identifying methodological and measurement gaps in vaccination timeliness studies (Box 1). Although conducted for different purposes compared to systematic reviews, scoping reviews still require rigorous and transparent methodologies to ensure that their results are trustworthy [13].

### Box 1. Potential measurement and methodological gaps in vaccination timeliness studies

*There are important issues that must be considered during data collection, analysis, and presentation of results in vaccination timeliness studies to ensure robustness and comparability of results. We refer to the key issues related to the collection of data and analysis as 'methodological gaps', while those related to how results are presented as 'measurement gaps'.*

#### Methodological gaps

1. How missing vaccination dates are handled: to effectively generate robust estimates for vaccination timeliness, precise vaccination dates are required. Inadequately handling missing dates is a potential gap

2. Definition of vaccination timeliness: to be able to compare results or generate point estimates from multiple studies, uniformity in defining vaccination timeliness is desirable.

#### Measurement gaps

1. Operationalisation of vaccination timeliness: how timeliness is reported or operationalized (continuous vs categorical) determines the usefulness of the estimates produced.

2. Domains of vaccination timeliness studied: domains of timeliness includes; 'early', 'untimely interval', or 'delayed' vaccination. focusing on one domain without the other is a potential measurement gap.

3. Determinants of vaccination timeliness: several factors act as barriers to receiving vaccines in a timely age-appropriate manner. Narrowly focusing on a few determinants could be considered a measurement gap.

This scoping review was based on the guidance framework of the Joanna Briggs Institute (JBI) [14]. The review is reported using the Preferred Reporting Items for Systematic Reviews and Meta-Analyses extension for Scoping Reviews (PRISMA-ScR) (S1 Checklist) [15]. Since registration of scoping reviews are currently not accepted in PROSPERO, we published the review protocol *a priori* in a peer-review journal [16]. The review process did not deviate from the previously published protocol.

## Search strategy

The literature search was performed across the databases: MEDLINE, EMBASE, Global Health, CINAHL and Web of Science. Following the recommendation of the JBI, we followed a three-step search strategy to ensure a comprehensive search [17]. First, a preliminary search of MEDLINE and Web of Science was conducted on March 27, 2021 using the key search concepts: *Childhood vaccination*; *Timeliness*; and *LMICs*. We refined the initial search strategy by including additional key concepts after analysing the text words in the title and abstract of the retrieved papers, and the indexing terms. The search strategy was developed in consultation with a Librarian and was refined based on their input. The full search strategy and search terms used in MEDLINE is included as S1 Table. In the second step, we conducted a full search on July 01, 2021, across all five included databases using the refined search strategy from the first step. The search strategy was adapted to fit the search terminologies for each database. In the third step, we searched the reference list of the included papers (from the database search) for additional sources not previously retrieved.

## Inclusion criteria

Studies were included if they reported childhood vaccinations that are part of the routine national EPI schedules; calculated any measure of timeliness related to vaccine coverage; are based on data from countries categorised as LMICs (low-income, lower middle-income, and upper-middle income economies) according to 2020 World Bank classification; [18] were published in English or French languages from 01 January 1978 through to July 01,2021. We restricted the review to studies conducted in LMICs because these countries account for a higher proportion of the global burden of VPDs, and the national EPI schedule in these countries generally adopts the WHO-recommended childhood immunization schedule. We did not include grey literature because it was unmanageable to manually search for additional official reports on vaccination timeliness from the EPI website of the more than 120 listed LMICs. We included studies published from 01 January 1978 because routine childhood immunization against diphtheria, pertussis, tetanus, poliomyelitis, measles, and tuberculosis commenced in LMICs in 1977 [1]. The search was extended to July 01,2021 to capture up-to-date evidence

on timeliness of routine childhood vaccination. We excluded systematic reviews, study protocols, journal commentaries, and conference papers.

## Study selection

Retrieved titles were imported into Endnote X9.3.3 (Clarivate Analytics) for de-duplication of records. Subsequently, the records were exported to Rayyan–a novel web based application for screening articles for reviews [19]. Two reviewers (OW and YKK) independently screened the titles and abstracts for relevance using the pre-set eligibility criteria. Records that met the eligibility criteria were exported back to Endnote for full-text retrieval, screening, and extraction. One reviewer (OW) screened the full text of records to ensure they were appropriate for full data extraction while another reviewer (YKK) verified all decisions. Final decisions regarding the eligibility of articles were made through consensus. A third member of the review team (UO) was consulted to resolve disagreements when the two initial reviewers fail to reach a consensus. All decisions were based on consensus.

**Data extraction.** We used a data extraction template to extract the information of interest from the included articles. We adapted the template from the JBI data extraction tool for scoping reviews [20]. Before the commencement of data extraction, two members of the review team piloted the extraction template on 20 randomly selected articles and was subsequently refined based on feedback from this process. One reviewer (YKK) extracted the data from the included articles while another reviewer (OW) verified the extracted data by cross-checking 10% of the full-text articles against the extracted data to ensure that the correct variables have been extracted. Critical appraisal of the quality of the included studies was deemed to be beyond the scope of this study and is not considered mandatory for scoping reviews [20].

## Presentation and charting of results

We analysed the extracted data descriptively and results are presented using tables, charts, and maps to ensure adequate visualisation of the key findings. We presented the number of studies published per country from 1978–2021 using a thematic map. The determinants of timeliness of routine childhood vaccination are organised according to *a priori* categories adapted from the 3-delays conceptual framework [21]. *Delay-1* relates to decision to seek care and includes factors such as household socioeconomic and cultural characteristics; *Delay-2* relates to arrival at a health facility and includes factors such as geographic accessibility and transportation; *Delay-3* are factors related to provision of adequate care at facility level [21]. We categorised the included studies to determine if censored data was accounted for during data analysis. Studies that statistically adjusted for children yet to be vaccinated at the time of the empiric studies are considered to have accounted for *right censoring.* On the other hand, studies that statistically adjusted for children vaccinated before data collection but without precise vaccination records are considered to have accounted for *left censoring.*

## Role of the funding source

The funder of the research had no role in the design, selection, data collection, data analysis, data interpretation, or writing of the report of this scoping review.

## Results

A total of 6 819 publications were identified (Fig 1). After duplicate removal, 4263 records were eligible for screening. After screening these records by title and abstract, 260 publications

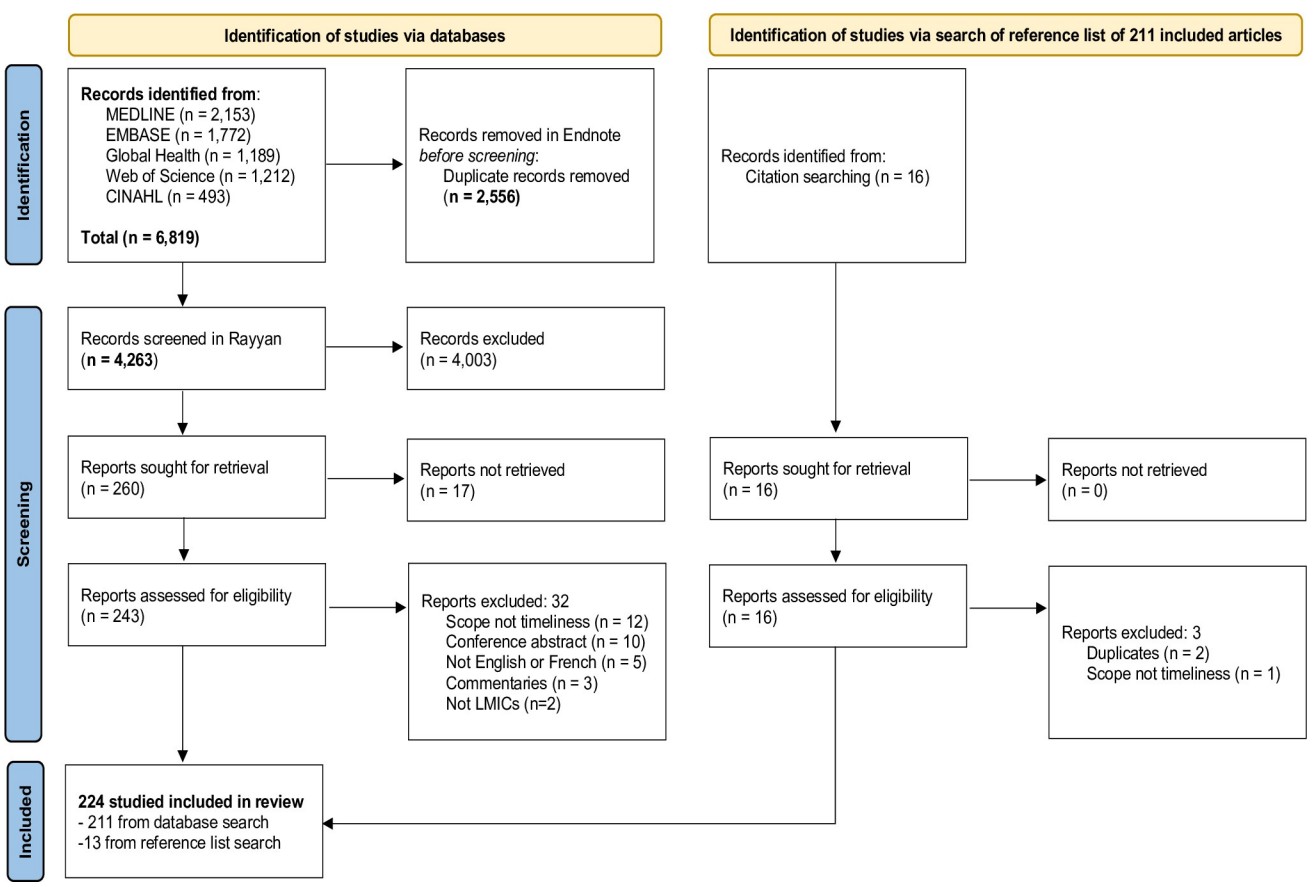

**Fig 1. Flowchart showing study identification, screening, and selection process.**

were selected for full-text screening; however, full-texts were not available for 17 titles even after contacting their authors as these articles were not open access. We further excluded 32 articles, leaving 211 articles for inclusion and 13 additional articles that fulfilled the inclusion criteria were identified from a search of the reference lists of the 211. Overall, 224 studies were included for analysis of which 13 were multi-country studies with the remaining 211 being single country reports. (S2 and S3 Tables).

Over one-third (35%; 78/224) of published studies were from the WHO African region, with only 2% (6/224) and 6% (15/224) from the European and America Region, respectively (Fig 2B). The included studies represented 103 of the 137 LMIC studied (S1 Table and S1 Checklist) with China (WHO Western Pacific Region; 40 articles), India (WHO South-East Asia Region; 27 articles), and Kenya (WHO African Region; 23 articles) being most represented countries (Fig 3).

The earliest reported study exploring timeliness of routine childhood vaccination in LMICs was published in 1987 [23]. Since 2004, we observed a gradual increase in relevant publications with the most rapid increase from 2013, with 20 articles already published in the first six months of 2021 (Fig 2A). The most common vaccines that have been the focus of studies on the timeliness of routine childhood are DTP3/Penta3 and MCV1 with 137 articles each. The least studied antigen was the yellow fever vaccine while the second doses of multi-dose vaccines were generally less studied (Fig 2C).

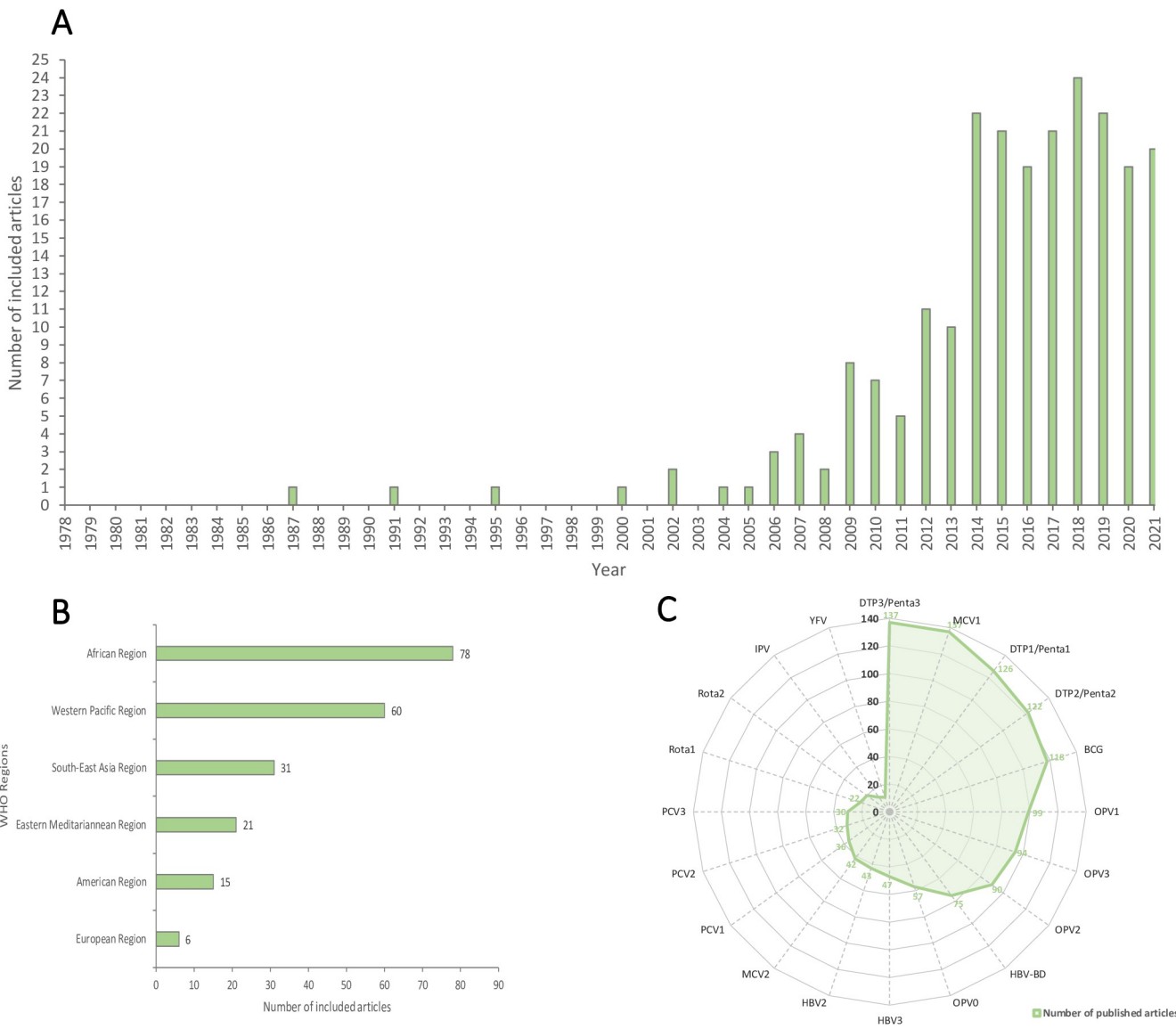

**Fig 2.** (a) How the literature on the timeliness of routine childhood vaccination has evolved, 1978–2021 Number of studies published per year (b) number of studies published per WHO region (c) antigens studied in the published literature.

### Domains and definitions of vaccination timeliness

All included studies but one [99.5% (223/224)] [24] explored the timeliness domain of 'delayed vaccination'. Less frequently studied were 'early vaccination' (receipt of a vaccine before the recommended schedule; 21.9% (49/224) of studies) and 'untimely interval vaccination' (receipt of a subsequent dose of a multi-dose antigen outside the recommended EPI window; 9% (20/224) of studies) (Fig 4A). We observed varying cut-off values for defining 'untimely interval', 'early', or 'delayed' vaccination. Among studies exploring 'untimely interval vaccination', four different definitions were used but over half [55% (11/20)] of the studies considered 4 weeks beyond the accepted EPI interval as being untimely (Fig 4B). Among the 49 studies that focused on 'early vaccination', seven different definitions were used, with the most used definition [63% (31/49)] being 'any time before the accepted EPI schedule' (Fig 4C). With 19

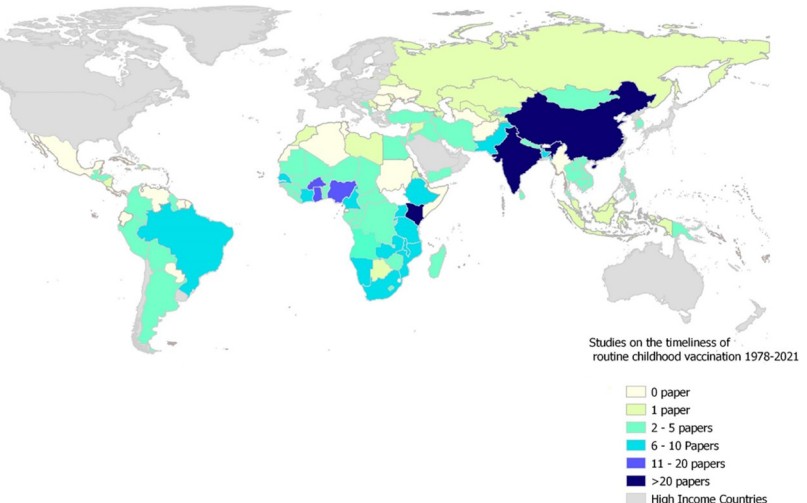

**Fig 3. Map of the world showing low-and middle-income countries where studies on the timeliness of routine childhood vaccination has been conducted, 1978–2021.** This map was produced by the authors with administrative boundaries data from geoBoundaries [22].

different definitions, delayed vaccination had the highest number of definitions of the domains studied (Table 1). Specifically, delayed birth-dose of hepatitis-B vaccine was defined in 15 different ways (Fig 4D).

## Operationalisation of vaccination timeliness

Untimely interval, early, and delayed vaccination were measured or operationalised in various ways by the included studies (Fig 4A). Most studies [72.3% (166/224)] operationalised untimely interval, early, and delayed vaccination as categorical measures such as the proportion of the study population with the different domains of vaccination timeliness using the operational definitions. However, only 9.8% (22/224) of studies operationalised these domains using continuous measures such as median and mean delay or early vaccinations (Fig 4A).

## Methodological and statistical gaps

During data collection for the included studies, the majority [47.8% (107/224)] excluded the data of children whose caregivers had no vaccination cards or written records of their vaccination irrespective of caregivers' recall of their vaccination status (Table 2). In 9.4% (21/224) of studies, it was not clear how scenarios where vaccination records were not available for some children was handled by the authors.

The majority [76.3% (171/224)] of studies did not account for any form of censored event [i.e., a child being vaccinated before their study but without a record (*left censoring*) or vaccination that would occur outside their study period (*right censoring*)]. There were 50 studies (22.3%) that accounted only for right censored data—i.e., children who were not vaccinated as of the time of the study but with a possibility of being vaccinated afterwards. Most of these studies used survival analysis techniques such as Kaplan-Meier statistics. Only three studies [98, 109, 147] accounted for both right and left censoring using survival analysis approach such as Turnbull and Weibull statistics (Table 2).

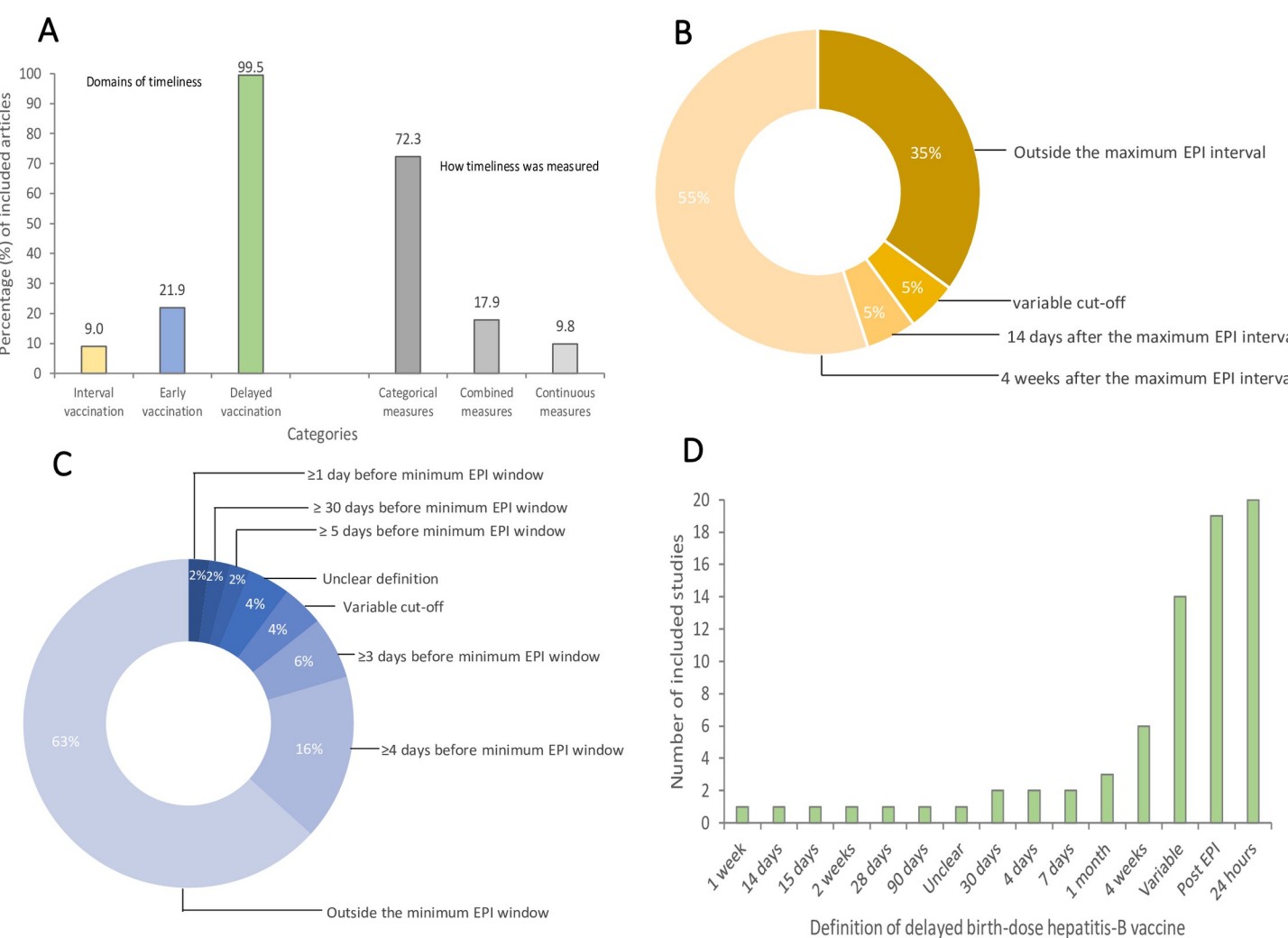

**Fig 4.** How the timeliness of routine childhood vaccination was defined and measured in the literature, 1978–2021 (a) domains of timeliness explored and how timeliness was operationalised (b) how untimely interval vaccination was defined (c) how early vaccination was defined (d) how delayed birth-dose hepatitis-B vaccine (HBV-BD) was defined. Note, in Fig 4D, the definition timelines are relative to the day of birth.

## Determinants of vaccine timeliness

Over two-thirds [68.3%; 153/224] of studies discussed factors associated with socioeconomic and household-level determinants (Delay 1) specifically, maternal education (46.4%); child's sex (35.3%); family wealth (33%); place of residence (29.5%); maternal age (27.2%); child's place of birth (25%); and maternal occupation (20.1%). Factors associated with accessibility of health facilities or immunisation clinics (Delay 2) were the least explored, accounting for 15.6% (35/224) of studies (Fig 5). Among the Delay 2 factors, reported travel distance was the most explored in the literature [35, 47, 59, 61, 65, 68, 77–79, 87, 116, 119, 131, 165, 168, 182, 203, 206, 217, 222, 223, 234, 235, 242]. Broader determinants such as conflict/humanitarian crises, and large public health crises such as COVID-19 which fall outside the traditional 3-delay categories, have been rarely studied. So far, only one published study has explored the impact of the ongoing COVID-19 pandemic on the timeliness of receiving routine childhood vaccination over an 18-month period following the onset of the pandemic (i.e., January 2020 – July 2021) [132].

**Table 1. How delayed routine childhood vaccination was defined in the 223 studies from LMICs that focused on this domain of timeliness, 1978–2021.**

| Studies | Cut-off or definition used | Definition of delayed vaccination |
|---|---|---|
| [25–47] | 24 hours | Hepatitis B vaccine birth doses received after 24 hours of birth |
| [48–50] | 4 days | EPI vaccine doses received ≥4 days after the recommended age of vaccination |
| [51–56] | 1 week or 7 days | EPI vaccine doses received 1 week or 7 days after the scheduled or recommended age of vaccination |
| [57–61] | 14 days or 2 weeks | EPI vaccine doses received 14 days or 2 weeks after the scheduled or recommended age of vaccination |
| [62] | 15 days | EPI vaccine doses received 15 days after the scheduled or recommended age of vaccination |
| [63–95] | 28 days or 4 weeks | EPI vaccine doses received 28 days or 4 weeks after the scheduled or recommended age of vaccination |
| [96] | 29 days | EPI vaccine doses received 29 days after the recommended age of vaccination |
| [97–106] | 30 days | EPI vaccine doses received 30 days after the recommended age of vaccination |
| [107, 108] | 30.5 days | EPI vaccine doses received 30.5 days after the recommended age of vaccination |
| [109] | 32 days | EPI vaccine doses received 32 days after the recommended age of vaccination |
| [110–121] | 1 month | EPI vaccine doses received 1 month after the scheduled or recommended age of vaccination |
| [122] | 2 months | EPI vaccine doses received 2 months after the recommended age of vaccination |
| [123] | 60 days | EPI vaccine doses received 60 days after the recommended age of vaccination |
| [124] | 90 days | EPI vaccine doses received 90 days after the recommended age of vaccination |
| [125] | >12 months of life | EPI vaccine doses received after 12 months of life |
| [23, 126–202] | Outside EPI window* | EPI vaccine doses received outside the country-specific EPI or WHO recommended vaccination windows |
| [203–205] | Outside manufacturer's recommended window | EPI vaccine doses received outside the manufacturer's recommended vaccination windows |
| [206–211] | Unclear cut-off** | Although delayed vaccination was studied, there was no clear definition or cut-off value |
| [212–245] | Variable cut-off*** | Several cut-offs used in the same study to define delayed vaccination of the same or different antigens in the schedule |

*These relied on the national EPI window in the country of the study. Any vaccine received outside the maximum date of the window was considered delayed.

**These studies focused on the domain 'delayed vaccination', however, did not explicitly document what operational definition was used.

***These studies measured delayed vaccination using multiple or variable definitions and reported multiple estimates for delayed vaccination.

## Discussion

Our scoping review show that 'delayed vaccination' was the commonest domain of vaccination timeliness studied, however, there were varying definitions for early, untimely interval, and delayed vaccination even in studies from the same country or focused on same vaccine. Most of the studies operationalised vaccination timeliness as a categorical variable. There was a lack of uniformity in handling situations where children were already vaccinated but lacked

**Table 2. Analytic and statistical gaps in the 224 included studies on the timeliness of routine childhood vaccination, 1978–2021.**

| Variable | Number of articles (N = 224) | Proportion (%) |
|---|---|---|
| **Statistically accounting for censored data** | | |
| Not done | 171 | 76.3 |
| Right censoring only | 50 | 22.3 |
| Both Right and Left censoring | 3 | 1.4 |
| **Unavailable precise vaccination records** | | |
| Excluded data | 107 | 47.8 |
| Not applicable* | 75 | 33.5 |
| Unclear | 21 | 9.4 |
| Included data | 21 | 9.4 |

*These studies were based on data from health information management systems (HIMS) or facility-based records, hence, vaccination dates were available.

information on precise vaccination dates. Demand-side factors such as socioeconomic and cultural determinants were most commonly studied, while supply-side or broader determinants such as factors related to accessibility of immunisation service points were the least studied determinants.

Vaccination schedules are designed with age-specific immunity and risks of disease in mind, thus, they target the best possible points of early childhood to ensure children develop

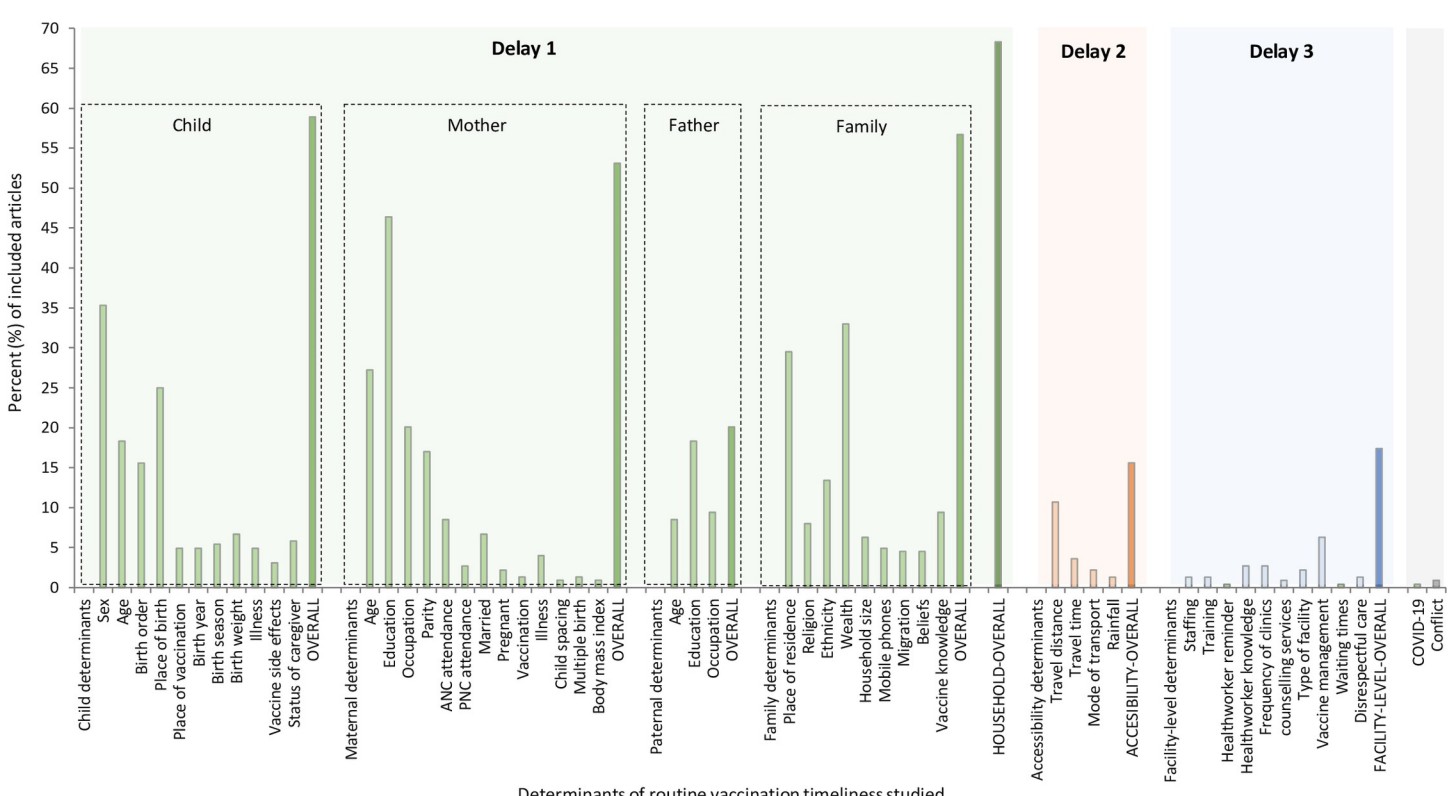

**Fig 5. Determinants of the timeliness of routine childhood vaccination studied in low-and middle-income countries, 1978–2021. Note**: Delay 1, 2, and 3 are based on the 3-delay conceptual framework developed by Thaddeus and Maine [21].

adequate immunity against VPDs as early as possible [7]. Furthermore, the intervals for multi-dose antigens is aimed at optimising immune responses against VPDs [5]. The vaccination schedules in early infancy, therefore, leaves little room for vaccination to be given before their due dates or delayed significantly. Although there are recommendation from the WHO regarding vaccination schedules, country-level vaccination windows are designed, taking into consideration, the local disease epidemiology, availability of resources, programmatic and policy considerations. Thus, the recommended age of vaccination for a specific vaccine in some LMICs might differ slightly from those in other countries. The lack of comparable definitions for early, untimely interval, and delayed vaccination could be partly explained by these variations in accepted windows across LMICs. However, we found that even for antigens such as the birth-dose of hepatis-B vaccine that is recommended within the $1^{st}$ 24 hours of life by the WHO, [246] there was no uniformity in the definitions used across studies. Irrespective of context, or antigen of focus, generating point estimates around each domain of vaccination timeliness through a meta-analysis to understand intra- or inter-country gaps in reaching antigen-specific targets will be limited due to the heterogeneity in the cut-off points used across studies.

An important measurement gap in the literature was that most studies operationalised the domains of vaccination timeliness as categorical variables. That is, most studies categorised doses as either 'on-time' if received within the cut-off points of the operational definition used, or as early, untimely interval, or delayed vaccination if received outside the specified operational definition, reported as proportions. While this approach appears pragmatic, it potentially lumps together a wide window of untimely vaccinations and obscures a nuanced interpretation of the data on vaccination timeliness. Unpacking and presenting the domains of vaccination timeliness as continuous variables, for example, as mean or median days delayed (outside the nationally accepted window) could be considered more robust. Clusters of children with a longer mean delay, potentially have a higher risk of VPDs exposure and likelihood of not completing their schedules compared to their counterparts with shorter delays. Additionally, comparatively longer untimely vaccination in a particular sub-national unit, potentially highlights equity gaps which must be bridged, or an early warning sign of weaknesses in immunisation programmes.

A key methodological gap was the lack of a uniform approach in handling censored data—i.e., situations where vaccination dates or time to vaccinations were not available for all participants. The commonest, in the included studies *left and right censoring*. Only three studies [98, 109, 147] accounted for both scenarios where precise vaccination dates were unavailable. Left censored data is common in LMICs where retention rates of vaccine cards are variable, and complete clinical records are seldom available. Using approaches that account for both right and left censoring improves the robustness of timeliness estimates, because it permits more observations (including those without vaccination records) to be included in the analysis that might otherwise have been excluded.

Factors related to the geographic accessibility of immunisation clinics, clinic-level and service delivery-related determinants have been less studied, compared to socioeconomic and cultural determinants of vaccination timeliness. This finding could have been due to the fact that data on geographic accessibility of health services is not routinely collected as part of health surveys, as this requires skilled personnel to collect and adequately model accessibility, compared to describing socioeconomic variables which are more routine. Nonetheless, there is evidence to suggest that geographic accessibility to immunization service points impacts the likelihood of receiving childhood vaccination in an age-appropriate manner [247, 248]. How remotely away from a clinic a family lives, how long they had to travel for an appointment,

geospatial relationships in catchment areas of clinics and the presence (or lack thereof) of accessible roads can all impact the uptake of health services, including vaccination.

There is no doubt that epidemics/pandemics, conflicts, and disasters such as earthquakes and flooding impact the delivery of health services, including timely receipt of vaccines. Since December 2019, an additional challenge has been posed by the ongoing COVID-19 pandemic, which has resulted in disruptions of immunisation systems [3, 249]. Despite the potential effect of the COVID-19 pandemic on routine vaccination timeliness, to date, only one study, [132] have explored the impact of the pandemic on vaccination timeliness. It is expected that the COVID-19 pandemic will continue to determine how timely children in many LMICs receive their vaccines, thus, future studies should explore its impact on vaccination timeliness. Understanding how, where, and to what extent fragile contexts impact the timeliness of receiving routine vaccination is an important initial step for EPI programmes to plan mitigating measures during such circumstances.

Our study has some limitations which must be considered. First, by including only studies published in English and French, we could have omitted a small number of studies published in other languages. Similarly, we did not include grey literature such as official government reports on vaccination timeliness. We also acknowledge that a handful of studies would have been published since our search was completed on 01 July 2021 as our study is not a 'living Review'. While a very small number of reports might have been published after we concluded our search, or might have been published in other languages and as grey literature, we do not expected them to significantly alter the conclusions drawn from our study which was based on 224 published articles, spanning 1978–2021. Second, we did not include studies that focused on vaccinations given outside the routine childhood EPI schedule, including those given in adolescence, and adulthood, for example maternal tetanus vaccinations. Third, although appraisal of study quality or design is primarily not the focus of scoping reviews, there was substantial variability in the quality and design of the included studies that potentially explains the observed measurement and methodological gaps. Despite these limitations, our study highlight important gaps related to the design, conduct and reporting of studies on vaccination timeliness that could shape future studies on this topic, and potentially improve their utility and comparability.

To date, this is the most extensive review spanning four decades aimed at understanding the measurement and methodological gaps in the literature on the timeliness of routine childhood vaccination in LMICs. To our knowledge, the first and only previous review on the subject by Masters et. al. (2019) [10] provided valuable insights into some existing measurement and methodological gaps in the literature on timeliness of vaccination; however, the review had key limitations that necessitated a further review. First, the review was limited to studies published between 2007–2017, and therefore did not include important studies published prior to 2007, or after 2017. Second, the review focused on three electronic databases and was restricted to studies published in English language only. Due to the extensive nature of our scoping review, we included 224 studies compared with only 67 in the review by Master et.al, thus, making our study more extensive.

## Implications for future research, policy, and practice

Based on our findings, future studies on the timeliness of routine childhood vaccination should, at minimum, pay attention to the following methodological and measurement issues to ensure the robustness, comparability, and utility of their findings. First, to bridge the methodological gap related to lack of a comparable cut-off or definition of early, untimely interval, and delayed vaccination, future studies should consider defining vaccine doses received

outside the nationally accepted EPI vaccination windows in their countries as early, untimely interval, or delayed as was done by some studies. Second, operationalising untimely vaccination as a categorical variable prevents a nuanced interpretation of vaccination timeliness. Thus, future studies should unpack and present the domains of vaccination timeliness as continuous variables, for example, as mean or median days vaccination was early or delayed outside the nationally accepted window. Through this approach, one can more clearly compare not just the proportion of children with untimely vaccination, but also on average, how many days outside the national vaccination window children are vaccinated too early or delayed across antigens–an important indicator of the quality of an immunisation programme. Also, such continuous variables can be easily converted to categorical variables, which may be more suitable when analysing individual level data. Third, deploying methodological approaches that account for situations where precise vaccination dates are unavailable potentially improves the power of the individual studies, thus, generating more reliable and precise estimates. Future studies can apply the Turnbull estimator, Weibull method, [250] or machine learning techniques to account for both left and right censored data as was done by three of the included studies. Fourth, to gain a robust understanding of the complex factors determining the timely receipt of vaccines, future studies should not only explore demand-side factors such socioeconomic or cultural determinants, but also, supply-side determinants including geographic accessibility to clinics, and facility-level factors. Lastly, the WHO and national immunisation programmes should develop and implement guidelines for measuring vaccination timeliness based on the accepted vaccination windows. Through this approach, measurement gaps related to the lack of a uniform cut-off for defining vaccination timeliness can be bridged, thus, improving the comparability and utility of data across antigens and settings.

## Supporting information

**S1 Table. Full search strategy in MEDLINE (Ovid).**
(DOCX)

**S2 Table. Summary characteristics of included studies.**
(DOCX)

**S3 Table. List of 46 low-and middle-income countries that were not the focus of a single study but contributed data to the 13 studies that were based on multiple countries.**
(DOCX)

**S1 Checklist. Preferred Reporting Items for Systematic reviews and Meta-Analyses extension for Scoping Reviews (PRISMA-ScR) checklist.**
(DOCX)

## Acknowledgments

We would like to acknowledge Russell Burke, Assistant Librarian, Library & Archives Service, at the London School of Hygiene and Tropical Medicine (LSHTM) for providing guidance during the development and refinement of the search strategy for this scoping review.

## Author Contributions

**Data curation:** Oghenebrume Wariri, Yakubu Kevin Kwarshak.

**Formal analysis:** Oghenebrume Wariri.

**Methodology:** Oghenebrume Wariri, Uduak Okomo, Yakubu Kevin Kwarshak, Chigozie Edson Utazi, Kris Murray, Chris Grundy, Beate Kampmann.

**Project administration:** Oghenebrume Wariri.

**Resources:** Oghenebrume Wariri.

**Supervision:** Uduak Okomo, Chigozie Edson Utazi, Kris Murray, Chris Grundy, Beate Kampmann.

**Visualization:** Oghenebrume Wariri.

**Writing – original draft:** Oghenebrume Wariri, Uduak Okomo.

**Writing – review & editing:** Oghenebrume Wariri, Uduak Okomo, Yakubu Kevin Kwarshak, Chigozie Edson Utazi, Kris Murray, Chris Grundy, Beate Kampmann.

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
