## [Decision Letter · Decision Letter 0]

21 Feb 2022

PGPH-D-21-01137

Timeliness of routine childhood vaccination in 103 low-and middle-income countries: a scoping review to map measurement and methodological gaps, 1978 – 2021

Dear Dr. Wariri,

Thank you for submitting your manuscript to PLOS Global Public Health. After careful consideration, we feel that it has merit but does not fully meet PLOS Global Public Health’s publication criteria as it currently stands. Therefore, we invite you to submit a revised version of the manuscript that addresses the points raised during the review process.

We look forward to receiving your revised manuscript.

Kind regards,

Meru Sheel

Academic Editor

Journal Requirements:

1. We note that the original search was performed in July 2021. Please discuss whether relevant literature has been published in the interim that would be expected to affect the results of the meta-analysis.

2. We have noticed that you have cited figures in the manuscript file. However, there are no corresponding files uploaded to the submission. Please ensure that all files are present to ensure that your paper is fully reviewed. Please provide separate figure files in .tif or .eps format only, and remove any figures embedded in your manuscript file. 

3. We have noticed that you have uploaded supporting information but you have not included a list of legends.  Please add a full list of legends for all supporting information files (including figures, table and data files) after the references list. 

Additional Editor Comments (if provided):

Reviewers' comments:

Reviewer's Responses to Questions

**Comments to the Author**

1. Does this manuscript meet PLOS Global Public Health’s publication criteria? Is the manuscript technically sound, and do the data support the conclusions? The manuscript must describe methodologically and ethically rigorous research with conclusions that are appropriately drawn based on the data presented.

Reviewer #1: Yes

Reviewer #2: Yes

Reviewer #3: Yes

2. Has the statistical analysis been performed appropriately and rigorously?

Reviewer #1: N/A

Reviewer #2: Yes

Reviewer #3: Yes

3. Have the authors made all data underlying the findings in their manuscript fully available (please refer to the Data Availability Statement at the start of the manuscript PDF file)?

Reviewer #1: Yes

Reviewer #2: Yes

Reviewer #3: Yes

4. Is the manuscript presented in an intelligible fashion and written in standard English?

Reviewer #1: Yes

Reviewer #2: Yes

Reviewer #3: Yes

5. Review Comments to the Author

Reviewer #1: PGPH-D-21-01137

The manuscript describes the methodological and measurement gaps in assessing the timeliness of routine childhood vaccination in low-income and middle-income countries (LMICs).

Very well written and clear abstract. The authors could use ‘low-and middle-income’ instead of ‘low-income and middle-income countries (LMICs)’ for consistency. The latter does not match with the title.

The author should follow the journal style to formate the manuscript.

Introduction

The author could discuss more on the study rationale. It does not answer the “So What” question adequately.

Methodology

‘We refined the initial search strategy after analysing the text words in the title and abstract’-How?. The author could elaborate a bit more on this so that the methodology can be replicated in future. What knowledge this study will add to the existing literature needs to be discussed.

The search was extended to 2021 to capture up-to-date evidence on the timeliness of routine childhood vaccination. The timeline is not clear. Earlier, you mentioned June 2021.

Single sentence paragraphs should be avoided.

Discussion

There are some repetitions of information, such as right and left censoring. The author should interpret this to not sound repetition of results in the discussion section.

Recommendations

The authors provided recommendations around methodological and programmatic issues; however, at the same time, the author may also discuss policy and program level factors that contributed to the timeliness and some recommendations around that.

For clarity, the author could use a PRISMA flow diagram.

Reviewer #2: # General:

Thank you for allowing me to review this interesting work. The authors did a good job mapping and summarising the published work on timeliness of routine childhood vaccination in LMICs; methods are sounds and the results are well displayed. I appreciate the methodological distinction made between ‘delayed’, ‘early’ and ‘untimely’ vaccination. I believe the work would benefit from taking the following points into account (further specified below):

First, I agree with the authors that work is particularly timely and relevant in the context of the COVID-19 pandemic affecting routine vaccination programmes worldwide (including in LMICs). However, the relationship between the pandemic and the timeliness of vaccination is not the aim of this paper and emphasising this link doesn’t do justice to the decades of literature (>200 studies!) that have been summarised in the work. I considered this work an important evidence base for future work investigating the impact of the pandemic on the timeliness of routine vaccination, rather than already measuring the impact itself. We are only just starting to scratch the surface of the impact of the pandemic on routine vaccination… The following recent publication reports on the first effects, and can be referenced in the discussion of this manuscript: https://doi.org/10.1016/S2214-109X(21)00512-X.

Second, regarding the methods, the choice for the current design and its methods (scoping review – still unknown to many) can be explained and justified with a bit more detail. Importantly, the decision to restrict to peer-reviewed work in English and French only needs to be well justified, if not reconsidered, as I believe including at least official reports from national (public health) institutes and inclusion of other languages would be a valuable addition to this work.

After revising the manuscript taking the comments presented below (and those of other reviewers) into account, I consider this work a valuable addition to the literature, of interest to the readership of PLOS GPH.

---

# Point-by-point:

* Key words: add ‘scoping review’

* Abstract - methods: specify “recognised scoping review approaches”

* Abstract: Revise the conclusion with something informative, as the current statement is very generic: “Our review provides evidence for key methodological gaps in the literature when reporting timeliness of childhood vaccination and develops recommendations that could shape the design and implementation of future research.”

* Abstract & Introduction – first paragraph: In line with my first general comments on COVID-19 and this work; please take the statement in the abstract “Despite its detrimental effect on uptake of childhood vaccination, to date, only one study published so far, has explored the potential impact of COVID-19 on vaccination timeliness.” out, because it is not surprising that not much has been published yet. The same applied for the first paragraph of the introduction: move the focus away from the pandemic and put the pandemic in the context of other factors contributing to timeliness of routine childhood vaccination (e.g. war, vaccine hesitancy).

* Introduction – last paragraph: please stick to the aim and objectives of the study and move the details to the method section of the work. Especially panel 1 belongs in the methods, not the introduction. Please add some argumentation for the chosen methods to help answer the research question: why did you decide to do a scoping review? Especially because this review method is close to the more well-known systematic literature review, it would be good to clarify this early (in the introduction), as well as in the methods.

* Methods – clarify and justify the choice of study design (scoping review). It is currently not clear how the study differs from its sibling-design, the systematic review.

* Methods - I very much appreciate the fact that the study protocol was published ahead of this submitted work, open access: https://doi.org/10.1371/journal.pone.0253423

* Methods - Reporting guidelines: the appropriate PRISMA-ScR checklist has been provided.

* Methods - I very much appreciate the acknowledgement of the PROSPERO registry, and fact that the study protocol was published ahead of this submitted work, open access: https://doi.org/10.1371/journal.pone.0253423

* Methods/discussion – grey literature: because not all countries and institutes report their data on vaccination timeliness in peer-reviewed literature (but in reports, e.g. by national public health institutes), I wonder to what extend data has been missed and why the authors decided to include peer-reviewed literature only? This should be discussed as a limitation of the review.

* Methods/discussion – languages: why published in English and French only? How many studies were excluded based on this language requirement? This should be discussed as a limitation of the review.

* Methods – typo/revise: this sentence doesn’t work “using a thematic using map.”

* Methods/results: it is not clear why a high number (17) of full-texts that were considered eligible based on title and abstract were not available (e.g. paywall). Did the authors contact the authors and ask them for the full text?

* Supplemental material: please number the respective supplements to make sure they are easy to find.

* Results: well presented!

* Discussion: well written overall.

* Discussion - limitations: Not necessarily limitations, but a choice made by design that can/should be discussed in the context of the study findings (but not as a limitation): „First, we did not include empiric studies from HICs which would have further strengthened our work.“ Please so add the language and literature restrictions as a limitation (see previous comment).

* Discussion: this recent publication might be relevant to add: https://doi.org/10.1016/S2214-109X(21)00512-X.

* Results/Discussion - context: I wonder if the authors can elaborate on the similarities and differences in the vaccine programmes itself; i.e. which vaccinations are given, when, and through which healthcare structures.

Reviewer #3: Dear Authors,

Many thanks for a very well written draft manuscript. This has been quite informative for me. While this is a scoping review of existing literature to map measurements and methodological gaps, I feel like one missing element was the mention of immunisation schedules and their role on this issue of timeliness.

I also wonder if you can strengthen the article a little more with a couple of sentences on how more documentation of timeliness outside of just the traditional metrics of performance would inform national policies on infant immunisations. Because the goal maybe to inform practise, but I think ultimately this type of documentation is more to provide country EPI managers with a reason to look into timeliness and the need to put in place policies that support timeliness for vaccinations.

Otherwise this is very informative, and definitely taps into an area that needs further exploration and refinement as the immunisation programs continue to evolve to meet the needs of the communities they serve.

I would be interested to see the maps and figures you have associated with the results once ready.

Kind wishes,

Sarah

6. PLOS authors have the option to publish the peer review history of their article (what does this mean?). If published, this will include your full peer review and any attached files.

**Do you want your identity to be public for this peer review?** For information about this choice, including consent withdrawal, please see our Privacy Policy.

Reviewer #1: **Yes: **Md Saiful Islam

Reviewer #2: No

Reviewer #3: **Yes: **Sarah W Wanyoike

---

## [Decision Letter · Decision Letter 1]

31 May 2022

PGPH-D-21-01137R1

Timeliness of routine childhood vaccination in 103 low-and middle-income countries, 1978 – 2021: a scoping review to map measurement and methodological gaps

Dear Dr. Wariri,

Thank you for submitting your manuscript to PLOS Global Public Health. After careful consideration, we feel that it has merit but does not fully meet PLOS Global Public Health’s publication criteria as it currently stands. Therefore, we invite you to submit a revised version of the manuscript that addresses the points raised during the review process.

We look forward to receiving your revised manuscript.

Kind regards,

Karen D. Cowgill, PhD, MSc

Academic Editor

Journal Requirements:

Additional Editor Comments (if provided):

Reviewers' comments:

Reviewer's Responses to Questions

**Comments to the Author**

1. If the authors have adequately addressed your comments raised in a previous round of review and you feel that this manuscript is now acceptable for publication, you may indicate that here to bypass the “Comments to the Author” section, enter your conflict of interest statement in the “Confidential to Editor” section, and submit your "Accept" recommendation.

Reviewer #1: (No Response)

2. Does this manuscript meet PLOS Global Public Health’s publication criteria? Is the manuscript technically sound, and do the data support the conclusions? The manuscript must describe methodologically and ethically rigorous research with conclusions that are appropriately drawn based on the data presented.

Reviewer #1: Yes

3. Has the statistical analysis been performed appropriately and rigorously?

Reviewer #1: Yes

4. Have the authors made all data underlying the findings in their manuscript fully available (please refer to the Data Availability Statement at the start of the manuscript PDF file)?

Reviewer #1: Yes

5. Is the manuscript presented in an intelligible fashion and written in standard English?

Reviewer #1: Yes

6. Review Comments to the Author

Reviewer #1: The paper “Timeliness of routine childhood vaccination in 103 low-and middle-income countries, 1978 – 2021: a scoping review to map measurement and methodological gaps” highlighted the methodological and measurement gaps in the assessment of timeliness

of routine childhood vaccination in low-income and middle-income countries. The paper might be considered for publication; however, I also think the paper requires minor revisions. The abstract should be revised to make it more informative. The rationale of the study is missing in the abstract. The abstract needs proofreading to avoid spacing mistakes. The abstract could be benefitted by adding one or two recommendations. In addition to the themes, the authors should add some data and revise the discussion of the abstract based on the data. In the body of the manuscript, the introduction and the methods are well developed. However, under the methods, how another reviewer verified the extracted data should be mentioned explicitly. How the conflicts were resolved should also be mentioned. Figure 4-D the timeline is a bit confusing and requires revisions.

7. PLOS authors have the option to publish the peer review history of their article (what does this mean?). If published, this will include your full peer review and any attached files.

**Do you want your identity to be public for this peer review?** For information about this choice, including consent withdrawal, please see our Privacy Policy.

Reviewer #1: **Yes: **Md Saiful Islam

---

## [Editor Report · Decision Letter 2]

14 Jun 2022

Timeliness of routine childhood vaccination in 103 low-and middle-income countries, 1978 – 2021: a scoping review to map measurement and methodological gaps

PGPH-D-21-01137R2

Dear Dr. Wariri,

We are pleased to inform you that your manuscript 'Timeliness of routine childhood vaccination in 103 low-and middle-income countries, 1978 – 2021: a scoping review to map measurement and methodological gaps' has been provisionally accepted for publication in PLOS Global Public Health.

Best regards,

Karen D. Cowgill, PhD, MSc

Academic Editor